# Emergent Area Operators in the Boundary

**Ronak M Soni**

*Chennai Mathematical Institute, H1, SIPCOT IT Park, Siruseri, Kelambakkam 603103, India*

*E-mail:* ronakmsoni@gmail.com

ABSTRACT: In some cases in two and three bulk dimensions without bulk local degrees of freedom, I look for area operators in a fixed boundary theory. In each case, I define an exact quantum error-correcting code (QECC) and show that it admits a central decomposition. However, the area operator that arises from this central decomposition vanishes. A non-zero area operator, however, emerges after coarse-graining. The expectation value of this operator approximates the actual entanglement entropy for a class of states that do not form a linear subspace. These non-linear constraints can be interpreted as semiclassicality conditions. The coarse-grained area operator is ambiguous, and this ambiguity can be matched with that in defining fixed-area states.

## 1 Introduction

One of the crucial themes that has emerged from the study of AdS/CFT has been *emergence*. The bulk is an approximate concept that emerges from the CFT in sufficiently classical states, and remembering that a state is not intrinsically dual to a single classical geometry has led to resolutions of information paradoxes [1–6]. One of the motivations of this work is to investigate this emergence in a Hamiltonian setting.

The specific example I will focus on is the emergence of an area operator that is expected to be dual to the modular Hamiltonian of the boundary CFT [7]. The modular Hamiltonian, when it exists, has a discrete spectrum, whereas its bulk dual, the area, is a continuous variable. This lack of discreteness is also related to an information paradox [1–3]. I will find that the origin of the continuous spectrum is coarse-graining, thus explaining a precise sense in which the area operator emerges. A similar (in spirit) coarse-graining appeared in [8] recently. The eigenstates of this emergent area operator will be identified with fixed-area states [9, 10].[1] The route I will take is to go back to the definition of the area operator in terms of quantum error-correction [12],[2] which I will call the 'information-theoretic area.'

---

[1] The fact that fixed-area states are coarse-grained objects has also been explored in [11], for example.
[2] See also [13–16] for generalisations and [17] for some simple examples

[12] showed that an exact quantum error-correcting code (QECC) with complementary recovery admits a central decomposition, and consequently that the entanglement entropy is given by an operator in the centre of the code algebra. I will identify specific QECCs (taking the cue from a proposal by [18]) in some holographic theories, and show that the central decomposition alluded to above exists. However, the consequent area operator vanishes.

I will then explain how an area operator emerges anyway after coarse-graining.[3] One of the hallmarks of this emergent area operator is that it calculates the entropy in a set of states that do not form a linear subspace. These non-linear conditions are characteristic also of a classical limit [19]. The fact that the area operator is emergent was also explained beautifully in [20], but that work did not put as much focus on coarse-graining.

Another motivation for this work was to understand the constructions of area operators in bulk terms [21–26] from defect operators or quantum subsemigroups, in purely boundary terms. While this exploration doesn't shed light on the emergence of the defect operator/quantum deformation per se, it does resolve an important technical problem with the cited works. In all of them, the entanglement was actually infinite due to the noncompactness of the group; from a coarse-graining perspective, this infinity is explicitly regulated.

## 2   Review

Let us begin with a short review of relevant results from the CFT and "quantum information and quantum gravity" literatures.

### 2.1   Holographic 2d CFTs

In this work, I will focus on theories dual to general relativity coupled to an $\mathcal{O}(1)$ number of matter fields at low energies. These holographic 2d CFTs have been studied intensely via bootstrap, see [27] for a pedagogical introduction. While we don't know sufficient conditions for a CFT to be holographic, some necessary conditions are known [28, 29]. A somewhat imprecise summary of these conditions is

1. The central charge $c = 3\ell/2G_N$ is large.

2. The theory has a normalisable vacuum.

3. The spectrum of primaries with $h, \bar{h} < c/24$ is sparse, i.e. the number of these primaries don't grow with $c$.

4. The theory has a twist gap, i.e. $\min(h, \bar{h}) > 0$ for any non-vacuum state.

Precise statements can be found in [27–29].[4]

A consequence of the above assumptions (minus the twist gap) is that the conformal weights of primaries above the black hole threshold, i.e. when $h, \bar{h} > c/24$, is dense. More

---

[3]I use emergence and coarse-graining as referring to essentially the same thing, but from some different perspectives. We coarse-grain a *description* to study emergent *phenomena*.

[4]The results in this work do not require the twist gap. The twist gap is required for the theory not to be 'stringy' in the bulk, as explained in [30] for example. I thank Suzanne Bintanja for explaining this to me.

precisely, an $\mathcal{O}(c^0)$ window contains $\mathcal{O}(e^c)$ operators [28, 29, 31, 32]. I will assume a slight extension of the results cited above in this work, which is that the number of Virasoro primaries in region of size $\delta \times \bar{\delta}$ in the $(h, \bar{h})$ plane is

$$\log \rho_{\mathrm{prim}}(h, \bar{h}) = 2\pi \sqrt{\frac{c-1}{6}} \left( \sqrt{h - \frac{c-1}{24}} + \sqrt{\bar{h} - \frac{c-1}{24}} \right) - \log c + \mathcal{O}(1). \qquad (2.1)$$

I will also assume that this formula holds in boundary CFT, with appropriate modifications [33, 34]. Proving this is beyond the scope of this work, and corrections to this statement will result in only minor (and straightforward) modifications below.

The coefficient of the log is consistent with what one finds after expanding the Virasoro modular S-matrix at large $c$ (keeping $h/c, \bar{h}/c$ fixed).[5] This assumption has been applied to holographic theories before, see for example [36, 37]. More precisely, the prediction for $\rho_{\mathrm{prim}}(h, \bar{h})$ one gets from the modular transformation of the vacuum character is

$$\rho_0(h, \bar{h}) = \frac{8}{P\bar{P}} \sinh 2\pi b P \sinh \frac{2\pi P}{b} \sinh 2\pi b \bar{P} \sinh \frac{2\pi \bar{P}}{b}, \quad \frac{c-1}{6} = b + \frac{1}{b}, \ P = \sqrt{h - \frac{c-1}{24}}. \tag{2.2}$$

and a similar relation between $\bar{h}, \bar{P}$. Taking the log and expanding at small $b \approx \sqrt{6/(c-1)}$ results in (2.1).

## 2.2 Areas: Geometric and Information-Theoretic

[12] considered an exact QECC $V: \mathcal{H}_{\mathrm{code}} \to \mathcal{H}_{\mathrm{phys}}$, such that $\mathcal{H}_{\mathrm{phys}} = \mathcal{H}_B \otimes \mathcal{H}_{\bar{B}}$. If there are two commuting algebras $a, a' \in \mathcal{L}(\mathcal{H}_{\mathrm{code}})$ such that $a$ $(a')$ is protected against erasure of $\bar{B}$ (B), then we say that the code has complementary recovery. The main result of [12] was that (up to some irrelevant details),

$$\mathcal{H}_{\mathrm{code}} \cong \bigoplus_\alpha \mathcal{H}_{b_\alpha} \otimes \mathcal{H}_{\bar{b}_\alpha}$$

$$|\psi\rangle = \sum_\alpha \sqrt{p_\alpha} |\psi_\alpha\rangle_{b_\alpha \bar{b}_\alpha}$$

$$\mathcal{H}_{\mathrm{phys}} \cong \bigoplus_\alpha \mathcal{H}_{B,\alpha} \otimes \mathcal{H}_{\bar{B},\alpha} \otimes \mathcal{H}_{\mathrm{fus},\alpha}, \quad \mathcal{H}_{\mathrm{fus},\alpha} = \mathcal{H}_{\mathrm{f},B,\alpha} \otimes \mathcal{H}_{\mathrm{f},\bar{B},\alpha}$$

$$V|\psi\rangle = \sum_\alpha \sqrt{p_\alpha} |\psi_\alpha\rangle_{B\alpha, \bar{B}\alpha} \otimes |\chi_\alpha\rangle_{\mathrm{fus},\alpha}. \tag{2.3}$$

Here, $\mathcal{H}_B = \oplus_\alpha \mathcal{H}_{B,\alpha} \otimes \mathcal{H}_{\mathrm{f},B,\alpha}$ and similarly for $\mathcal{H}_{\bar{B}}$. $|\chi_\alpha\rangle$ is independent of the code state $|\psi\rangle$. The $\alpha$ sectors are joint eigenstates of the centre $z = a \cap a'$, and so this is known as a central decomposition.

This results in a central decomposition of the entanglement entropy

$$S_E(B; V|\psi\rangle) = \langle \hat{A} \rangle_\psi + \sum_\alpha p_\alpha (-\log p_\alpha + S_E(B_\alpha; \psi_\alpha)), \qquad \hat{A} = \oplus_\alpha S_E(\mathcal{H}_{\mathrm{f},B,\alpha}; \chi_\alpha) \hat{P}_\alpha, \quad (2.4)$$

---

[5]It also agrees with an extrapolation of the Virasoro primary density of states in [32] (evaluated at $h, \bar{h} \gg c$) to the regime $h, \bar{h} \sim c$. It disagrees with the log correction in the full microcanonical entropy, where the coefficient is 3 [35].

where $\hat{P}_\alpha$s are projectors onto the corresponding $\alpha$ sectors. Note that the final formula is evaluated in $\mathscr{H}_{\text{bulk}}$ and all the information about $V$ is packaged into $\hat{A}$. The $\hat{A}$ operator is what I will call the information-theoretic area operator, since it is defined entirely in terms of QEC.

The hypothesis is that this information-theoretic area operator is dual to the geometric area operator. This hypothesis has withstood some non-trivial checks, see e.g. [9, 10, 20], and so it is common to ignore the distinction and just refer to $\hat{A}$ as the area operator. However, they are conceptually distinct things.

# 3 An Exact QECC

With the preliminaries out of the way, let us turn to actually implementing the framework of [12] in holographic 2d CFT. In section 3.1, I define a code subspace of states in $\mathscr{H}_{\text{CFT}}^{\otimes 2}$; a central decomposition is then found for general CFTs in section 3.2. The final result of this section is that the information-theoretic area operator introduced in [12] vanishes for this code.

## 3.1 Code Subspace

The code subspace of interest contains all states in $\mathscr{H}_{\text{CFT}}^{\otimes 2}$ that are, in a sense, 'descendants' of the infinite-temperature thermofield double state. More precisely, consider a TFD of some high temperature $\epsilon$,

$$|\epsilon\rangle \equiv \frac{1}{\sqrt{Z(\epsilon)}} \sum_E e^{-\frac{\epsilon}{2}E} |E\rangle_L |E\rangle_R. \tag{3.1}$$

The code subspace contains all states obtained by functions of the stress tensor acting on L,

$$\mathscr{H}_{\text{code}} = \{f(T_{\mu\nu} \otimes \mathbb{1}) |\epsilon\rangle\} \cap \mathscr{H}_{\text{CFT}}^{\otimes 2}. \tag{3.2}$$

This code subspace includes all thermofield double states, since $\exp\{-\beta H_L\}$ is a function of the stress tensor. Another class of states it includes is time-evolved TFD states, since $\exp\{iHt\}$ is also a function of the stress tensor. It also include states prepared by Euclidean tubes (manifolds of cylinder topology with non-constant radius). The final example of states in this code subspace is the states discussed in [38] — thermofield double states with descendant/boundary graviton excitations.

## 3.2 A Vanishing Area Operator

The Hilbert space of a CFT is given by[6]

$$\mathscr{H}_{\text{CFT}} = \bigoplus_{(h,\bar{h})\in\text{spec CFT}} \mathcal{V}_h \otimes \mathcal{V}_{\bar{h}} \equiv \bigoplus_{(h,\bar{h})} \mathscr{H}_{(h,\bar{h})},$$

$$\mathscr{H}_{\text{CFT}} \supset |\Psi\rangle = \sum_{(h,\bar{h})} \sum_{\mathbf{m},\bar{\mathbf{m}}} \Psi(h,\bar{h},\mathbf{m},\bar{\mathbf{m}}) |h,\bar{h};\mathbf{m},\bar{\mathbf{m}}\rangle. \tag{3.3}$$

---

[6]I am assuming that the CFT has no degeneracies, so that there is a unique primary with a given $h, \bar{h}$. Lifting this assumption is straightforward and only requires uninteresting notation.

Here, $\mathbf{m}, \bar{\mathbf{m}}$ are decreasing lists of natural numbers, which denote an orthonormal subspace for the corresponding Verma module, denoted by $\mathcal{V}$.[7] For a generic irrational CFT with only Virasoro symmetry, the only Verma module with null states is the identity module $\mathcal{V}_0$ and all the other descendant Hilbert spaces $\mathcal{H}_{(h,\bar{h})}$ are thus isomorphic in a natural way. As a result, we can write

$$\mathcal{H}_{\mathrm{CFT}} = \mathcal{H}_{\mathbb{1}} \oplus \mathcal{H}_{\mathrm{prim}} \otimes \mathcal{H}_{\mathrm{desc}}. \tag{3.4}$$

The Hilbert space of two CFTs is

$$\mathcal{H}_{\mathrm{CFT}}^{\otimes 2} = \bigoplus_{(h,\bar{h}),(h',\bar{h}')} \mathcal{H}_{(h,\bar{h})}^{(\mathrm{L})} \otimes \mathcal{H}_{(h',\bar{h}')}^{\mathrm{R}}, \tag{3.5}$$

where the primaries on the two sides are uncorrelated. The code subspace is

$$\mathcal{H}_{\mathrm{code}} = \bigoplus_{(h,\bar{h})} \mathcal{H}_{(h,\bar{h})}^{(\mathrm{L})} \otimes \mathcal{H}_{(h,\bar{h})}^{(\mathrm{R})} = \mathcal{H}_{\mathbb{1}}^{(\mathrm{L})} \otimes \mathcal{H}_{\mathbb{1}}^{(\mathrm{R})} \bigoplus \mathcal{H}_{\mathrm{prim}} \otimes \mathcal{H}_{\mathrm{desc}}^{(\mathrm{L})} \otimes \mathcal{H}_{\mathrm{desc}}^{(\mathrm{R})}, \tag{3.6}$$

where the same primary appears on both sides in the first expression. The proof of this statement is straightforward. Consider the wavefunction of any state $|\Psi\rangle \in \mathcal{H}_{\mathrm{code}}$ in the basis

$$\left| h, \bar{h}; \mathbf{m}, \bar{\mathbf{m}} \right\rangle \propto \underset{L_{-\mathbf{m}}\bar{L}_{-\bar{\mathbf{m}}}O_{h,\bar{h}}}{\bigcup} . \tag{3.7}$$

This equation is the usual statement of the operator $\to$ state map in CFT, in a Weyl frame where the disk becomes a hemisphere. The corresponding path integral construction for $|\Psi\rangle$ is (somewhat schematically) a cylinder with stress tensor insertions. The wavefunction in this basis is

$$\Psi(h, \bar{h}, h', \bar{h}'; \mathbf{m}, \bar{\mathbf{m}}; \mathbf{m}', \bar{\mathbf{m}}') = \left( \langle h, \bar{h}; \mathbf{m}, \bar{\mathbf{m}} | \langle h', \bar{h}'; \mathbf{m}', \bar{\mathbf{m}}' | \right) | \Psi \rangle$$

$$= \quad \text{with various stress tensor insertions}$$

$$\propto C_{(h,\bar{h}),(h',\bar{h}'),\mathbb{1}} = \delta_{h,h'}\delta_{\bar{h},\bar{h}'}. \tag{3.8}$$

(3.6) is merely a compact expression of the fact that this is true for all $h, \bar{h}$.

This means that the reduced density matrix on the right subsystem takes the form

$$\rho_{\mathrm{R}} = \bigoplus_{(h,\bar{h})} p_{(h,\bar{h})}\rho_{(h,\bar{h})}, \quad \mathrm{tr}\,\rho_{(h,\bar{h})} = 1, \tag{3.9}$$

---

[7]An example construction is this. Consider the state $\prod_{m_i \in \mathbf{m}} L_{-m_i} \prod_{\bar{m}_i \in \bar{\mathbf{m}}} L_{-\bar{m}_i} |h, \bar{h}\rangle$ and perform a Gram-Schmidt orthogonalisation procedure.

and therefore that the entanglement entropy takes the form

$$S_E(R; |\Psi\rangle) = H(p_{(h,\bar{h})}) + \sum_{(h,\bar{h})} p_{(h,\bar{h})} S_{vN}(\rho_{(h,\bar{h})}), \tag{3.10}$$

where $H$ denotes the Shannon entropy. Note that both terms are non-linear in the state and therefore that no area operator has emerged. An independent but related observation is that this is true for a class of CFTs much larger than just holographic CFTs. The emergence of the area operator is a signature of the semiclassical limit, which we have not yet taken.

The connection between the lack of semiclassicality and the lack of an area operator can be seen as follows. The area operator measures the 'background' entanglement present in every state of the code, and any subspace containing all time-evolved thermofield double states also contains factorised states. Calling the thermofield double at inverse temperature $\beta$ and time-evolved by a time $t$ $|\beta + it\rangle$, we can write down

$$|E\rangle_L |E\rangle_R \propto \int dt\, e^{-iEt} \sum_{E'} e^{-\beta E' + iE't} |E'\rangle_L |E'\rangle_R \propto \int dt\, e^{-iEt} |\beta + it\rangle. \tag{3.11}$$

Since degeneracies are non-generic in holographic CFTs, this inverse Fourier transform picks out a factorised state for generic values of $E$ in the spectrum.[8] A superposition like this, which involves a super-exponential (in $c$) number of states, is not semiclassical. Therefore we have to impose semiclassicality by hand.

## 4 Emergence of an Area Operator from Coarse-Graining

To go further and bring (4.18) into the RT form in [12], some approximations are needed. This section, which is the central one in this work, concerns these approximations. These approximations will only be true when a (non-linear) set of conditions is satisfied.

The analysis here, unlike in section 3, will require the full set of assumptions outlined in section 2.1 (apart from the twist gap). Henceforth, I will replace the cumbersome notation $(h, \bar{h})$ for the primaries with $O$, and ignore the anti-holomorphic descendants (except where they are important). I will treat $O$ as both an abstract operator label and also a label for a discrete set of points in $\mathbb{R}^2$ with coordinates given by $(h, \bar{h})$.

A non-zero area operator might be found in a more sophisticated code, for instance one that includes time-evolved thermofield doubles only for $t \ll \mathbb{O}(e^c)$. But I will follow a different route, and find that an area operator emerges for a set of states that do not close under linear superposition.

The main idea is that, even though each primary sector does not have any background entanglement, primaries always appear in exponentially large 'bunches' in semiclassical states. I will argue that the $\mathcal{H}_{prim}$ in (3.5) effectively splits into

$$\mathcal{H}_{prim} \approx \oplus_\alpha \mathcal{H}_\alpha, \quad \text{and} \quad \rho_R \approx \oplus_\alpha p_\alpha \frac{\mathbb{1}_\alpha}{|\mathcal{H}_\alpha|} \tag{4.1}$$

---

[8]For generic $E \in \mathbb{R}$, the inverse Fourier transform vanishes, since the spectrum is discrete. If there are $\mathbb{O}(1)$ degeneracies, the state we obtain has $\mathbb{O}(1)$ entropy, but this is not the area operator we are looking for.

for $\rho_R$ the reduced density matrix of a semiclassical state. This *approximate* form will lead to an *emergent* area term in the entanglement.

I first show how we can go about coarse-graining a general underlying set to find an emergent area-like contribution in the entropy in section 4.1 and then turn to holographic CFTs in section 4.2 to argue that holographic CFTs have an emergent area operator.

## 4.1 Coarse-Graining Entropies: A General Discussion

Let us begin by considering a classical probability distribution over a discrete set $\mathfrak{O}$, $\{p_O | O \in \mathfrak{O}\}$. In this section, I will look at various ways to 'bin' this set and see how the Shannon entropy

$$H(p_O) = - \sum_{O \in \mathfrak{O}} p_O \log p_O \tag{4.2}$$

behaves under this coarse-graining. Note that this Shannon entropy is also the von Neumann entropy of a diagonal density matrix $\rho = \sum_O p_O |O\rangle \langle O|$.

One physical picture the reader can keep in mind is that there is an apparatus measuring $O$ that is not able to distinguish individual values of $O$; the different ways of binning below are different models for this finite-resolution apparatus. However, I emphasise that the mathematical expressions constitute a change of *description*. The two pictures are related in an active-passive sense: we can either keep the fine-grained description and also *model the apparatus*, or forget about the apparatus and *coarse-grain the description*.

For the simplest case, assume that $\mathfrak{O} = \bigcup_\alpha \mathfrak{O}_\alpha$ such that $p_O$ is a constant function on $\mathfrak{O}_\alpha$: $\forall O \in \mathfrak{O}_\alpha$, $p_O = p_\alpha/n_\alpha$, where $n_\alpha \equiv |\mathfrak{O}_\alpha|$. This is illustrated in figure 1. The Shannon entropy is

$$H(p_O) = - \sum_\alpha \sum_{O \in \mathfrak{O}_\alpha} \frac{p_\alpha}{n_\alpha} \log \frac{p_\alpha}{n_\alpha} = \sum_\alpha p_\alpha \log n_\alpha - \sum_\alpha p_\alpha \log p_\alpha, \tag{4.3}$$

where $p_\alpha \equiv n_\alpha q_\alpha$ is the probability of the sector $\alpha$. This Shannon entropy is the von Neumann entropy of a density matrix

$$\rho = \bigoplus_\alpha p_\alpha \frac{\mathbb{1}_{n_\alpha}}{n_\alpha}, \tag{4.4}$$

in which case (4.3) can be written as

$$H(p_O) = \left\langle \oplus_\alpha \log n_\alpha \hat{P}_\alpha \right\rangle_\rho + H(p_\alpha). \tag{4.5}$$

This first term is the expectation value of a linear operator. There is a linear subspace of states of the form (4.4) and the linear operator here is the information-theoretic area operator for this subspace. This is essentially the case dealt with in [12].

Let's add one more complication. Take again $\mathfrak{O} = \bigcup_\alpha \mathfrak{O}_\alpha$, but let's drop the assumption that the probability distribution is flat in each bin. Assume that $p_O = p_\alpha/n_\alpha + \delta p_{\alpha,O}$, such that $\sum_{O \in \mathfrak{O}_\alpha} \delta p_{\alpha,O} = 0$ and

$$\sum_{O \in \mathfrak{O}_\alpha} |\delta p_{\alpha,O}| \ll p_\alpha. \tag{4.6}$$

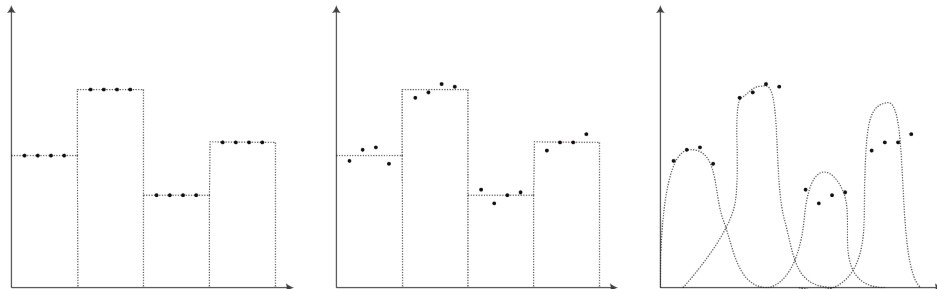

**Figure 1**: Successively more sophisticated coarse-graining schemes, from left to right. In the first, we assume that the fine-grained probability is constant within some bins. Secondly, the data varies but we average over a bin anyway. Finally, the bins become fuzzy also.

We find now

$$
\begin{aligned}
H(p_O) &= -\sum_\alpha \sum_{O \in \mathfrak{O}_\alpha} \left( \frac{p_\alpha}{n_\alpha} + \delta p_{\alpha,O} \right) \log \left( \frac{p_\alpha}{n_\alpha} + \delta p_{\alpha,O} \right) \\
&= \sum_\alpha \left[ p_\alpha \log \frac{n_\alpha}{p_\alpha} - \frac{1}{2} \sum_{O \in \mathfrak{O}_\alpha} \frac{\delta p_{\alpha,O}^2}{p_\alpha/n_\alpha} + \mathcal{O}\left( \delta p_{\alpha,O}^3 \right) \right].
\end{aligned}
\tag{4.7}
$$

The first term is again an area term, but this time an *approximate* one, for states $\rho = \sum_O p_O |O\rangle \langle O|$ that satisfy (4.6). Defining the projector $P_\alpha \equiv \sum_{O \in \mathfrak{O}_\alpha} |O\rangle \langle O|$, (4.6) can be written as

$$
\sum_O \left| \delta p_{\alpha,O} \right| \equiv \sum_{O \in \mathfrak{O}_\alpha} \sqrt{ \left[ \langle O|\rho|O\rangle - \frac{\operatorname{tr} P_\alpha \rho}{n_\alpha} \right]^2 } \ll \operatorname{tr} P_\alpha \rho.
\tag{4.8}
$$

Thus, (4.6) is non-linear in the state, and states that satisfy this do not form a subspace.

In both of these examples, the coarse-grained value was a fixed function of the fine-grained value. In general, however, one expects that if one feeds a fixed fine-grained state $O$ into a low-resolution instrument, it can spit out multiple values $\alpha$ with various probabilities. To model this, consider a generalised bin set given by a partition of unity $f_\alpha(O)$, satisfying $\int d\mu(\alpha) f_\alpha(O) = 1$. Here, $d\mu(\alpha)$ is some measure on the set of bins; the cases of continuous bin sets and discrete bin sets (or sets that are partly continuous and partly discrete) can be dealt with together using different choices of this measure. Remember that the underlying set $\mathfrak{O}$ is discrete by definition, so we are allowing for the coarse-grained variable to take on *more* values than the fine-grained variable. This is actually quite common: when we look at a particle on a one-dimensional lattice from very far away, it looks like a particle on a line, so we are coarse-graining the discrete lattice position into the continuous coordinate.[9]

$f_\alpha(O)$ is the probability that the instrument readout is $\alpha$ when it measures the state $O$. More generally, there is a stochastic map from $\mathfrak{O}$ to the set of bins, and $f_\alpha(O)$ is the probability that $O$ maps to $\alpha$. In other words, it is the *conditional probability* $p(\alpha|O)$. There is then also a *joint* probability distribution $p(\alpha, O) = p_O p(\alpha|O) = p_O f_\alpha(O)$ and also the marginal on $\alpha$, $p_\alpha = \sum_{O \in \mathfrak{O}} p_O f_\alpha(O)$. The Shannon entropy $H(O)$ can be rewritten using

---

[9]Of course, no actual instrument gives real-valued results. The lab example is merely for intuition.

the Bayes' rule,

$$H(O) = H(\alpha) + H(O|\alpha) - H(\alpha|O), \tag{4.9}$$

where the conditional entropies are defined as

$$H(\alpha) = -\int d\mu(\alpha)\, p_\alpha \log p_\alpha$$

$$H(O|\alpha) = -\sum_{O \in \mathfrak{O}} \int d\mu(\alpha)\, p(\alpha, O) \log \frac{p(\alpha, O)}{p_\alpha}$$

$$H(\alpha|O) = -\sum_{O \in \mathfrak{O}} \int d\mu(\alpha)\, p(\alpha, O) \log \frac{p(\alpha, O)}{p_O} = \mathbb{E}_{\mathfrak{O}}\left[-\int d\mu(\alpha)\, f_\alpha(O) \log f_\alpha(O)\right]. \tag{4.10}$$

Note that $H(\alpha)$ is by itself not a true entropy when the bin set is continuous (it is sometimes called a differential entropy); but the sum is well-defined and is a true entropy.

Of these terms, the most interesting is $H(O|\alpha)$. It is the uncertainty in the fine-grained variable given the coarse-grained one. Define again $p_\alpha, n_\alpha, \delta p_{\alpha,O}$ by the relations

$$p(\alpha, O) = \frac{p_\alpha}{n_\alpha} + \delta p_{\alpha,O}, \quad n_\alpha \equiv \sum_O f_\alpha(O), \quad \sum_O \delta p_{\alpha,O} = 0. \tag{4.11}$$

*Assuming* that the second term is small,

$$H(O|\alpha) = -\sum_O \int d\mu(\alpha) \left(\frac{p_\alpha}{n_\alpha} + \delta p_{\alpha,O}\right) \log\left[\frac{1}{n_\alpha}\left(1 + \frac{\delta p_{\alpha,O}}{p_\alpha/n_\alpha}\right)\right]$$

$$\approx \int d\mu(\alpha)\, p_\alpha \log n_\alpha + \mathfrak{O}\left[\left(\frac{\delta p_{\alpha,O}}{p_\alpha/n_\alpha}\right)^2\right]. \tag{4.12}$$

Considering again a quantum state $\rho = \sum_O p_O |O\rangle\langle O|$, this is the expectation value of the operator

$$\hat{A} \equiv \int d\mu(\alpha) \log n_\alpha \sum_{O \in \mathfrak{O}} f_\alpha(O) |O\rangle\langle O|. \tag{4.13}$$

Thus, for a given choice of bin functions, we find that

$$H(O) \approx H(\alpha) + \langle \hat{A} \rangle, \tag{4.14}$$

as long as

$$H(O|\alpha) - \langle \hat{A} \rangle \ll H(O) \quad \text{and} \quad H(\alpha|O) \ll H(O). \tag{4.15}$$

These conditions are (a) non-linear and (b) fuzzy (since they involve $\ll$). I will refer to these as semiclassicality conditions.

**The quantum entropy term**  The next question is how to extend this coarse-graining discussion to include the last, 'quantum,' term in (2.4). To do so, define

$$\rho_\alpha = \sum_O f_\alpha(O)\rho_O, \tag{4.16}$$

and impose another semiclassicality condition[10]

$$-\int d\mu(\alpha) \, p_\alpha \rho_\alpha \log \rho_\alpha \approx -\sum_O p_O \rho_O \log \rho_O. \tag{4.17}$$

With this, we find that the entanglement entropy can be written as

$$S_E(R;\Psi) \approx \langle \hat{A} \rangle + H(p_\alpha) + \int d\mu(\alpha) \, p_\alpha S_{vN}(\rho_\alpha), \quad \text{given (4.16) and (4.17).} \tag{4.18}$$

To apply this formalism to specific cases, we mainly need to specify

1. The space $\mathfrak{O}$.

2. The set of window functions and a calculation of $\mathfrak{n}_\alpha$.

3. And, finally, a reason to expect (4.15) and (4.17) to be satisfied.

## 4.2 Coarse-Graining the Entropy in Holographic CFTs

In the case of a holographic CFT, the set $\mathfrak{O}$ consists of a discrete set of primaries. For $h, \bar{h} = \mathcal{O}(c^0)$, the set is sparse. In this regime, we can take the set of $\alpha$s to agree with $\mathfrak{O}$ and take $f_\alpha(O) = \delta_{\alpha,O}$. So, $\mathfrak{n}_\alpha = 1$ in this regime.

In the regime where the spectrum is dense, take the bin functions $f_\alpha(O)$ to be functions with spread $\delta, \bar{\delta}$ on the $(h, \bar{h})$ plane. As discussed in section 2.1, the log of the number of primaries in this region is given by (2.1). This is also the leading order contribution to $\log \mathfrak{n}_\alpha$; since $f_\alpha(O)$ has a positive $\mathcal{O}(1)$ lower-bound as well as an $\mathcal{O}(1)$ upper-bound within this window, $\mathfrak{n}_\alpha$ receives $\mathcal{O}(1)$ multiplicative corrections from the non-trivial profile of $f_\alpha$, which becomes $\mathcal{O}(1)$ additive corrections in $\log \mathfrak{n}_\alpha$.

Plugging these values into (4.13) essentially gives the main result of this paper. The one missing ingredient is to rewrite it in terms of more familiar operators of 2d CFT. To do so, we use a Virasoro Casimir defined in [39]. One of the Casimirs defined there, which I will call $C_{Vir}$, has the property that for any descendant state $\left| L_{-\mathbf{n}} \bar{L}_{-\bar{\mathbf{n}}} O \right\rangle$,

$$C_{Vir} \left| L_{-\mathbf{n}} \bar{L}_{-\bar{\mathbf{n}}} O \right\rangle = h_O \left| L_{-\mathbf{n}} \bar{L}_{-\bar{\mathbf{n}}} O \right\rangle. \tag{4.19}$$

There is similarly an antiholomorphic operator $\bar{C}_{Vir}$ whose eigenvalues are $\bar{h}_O$. In [39], this operator is written down in terms of the Virasoro operators as an infinite series, but we only need that it satisfies (4.19). Remembering our no-degeneracy assumption, we can define the projector $P_O$ as the projector onto the $(h_O, \bar{h}_O)$ eigenspace of the 2-tuple-valued operator $(C_{Vir}, \bar{C}_{Vir})$. Below, I refer to this tuple of operators as $C_{Vir}$ for simplicity of notation. One perspective on this operator is as follows; there is a natural projector $|O\rangle\langle O| \in B(\mathcal{H}_{prim})$. (3.4) defines an isometric embedding $\mathcal{B} : \mathcal{H}_{prim} \hookrightarrow \mathcal{H}_{CFT}$,[11] and $C_{Vir} = \mathcal{B}|O\rangle\langle O|\mathcal{B}^\dagger$ is the image of this projector under this isometric embedding.

---

[10]This simple-minded procedure will be enough in the limit of interest below, where we will drop $G_N^0$ corrections. Going to higher orders will require a more sophisticated coarse-graining.

[11]The notation $\mathcal{B}$ is inherited from that for OPE blocks in [40].

Define spec$_-$ $C_{Vir}$ as the set of states below the black hole threshold, where the density of states (2.1) is not valid. This allows us to write down the final expression of an area operator in holographic 2d CFT for the code (3.6).

$$\hat{A} \equiv \int d\mu(\alpha) \log n_\alpha \sum_O f_\alpha(O) P_O, \qquad (4.20)$$

where

$$O \in \text{spec}\, C_{Vir}\,,$$

$$\alpha \in \text{spec}_-\, C_{Vir} \bigcup \left(\frac{c}{12}, \infty\right) \times \left(\frac{c}{12}, \infty\right),$$

$$f_\alpha(O) = \begin{cases} \delta_{\alpha,O} & \alpha \in \text{spec}_-\, C_{Vir} \\ \text{a function peaked at } \alpha = O \text{ and having } \mathcal{O}(1) \text{ width} & \text{else} \end{cases}$$

$$\text{and} \quad \log n_\alpha \approx \begin{cases} 0 & \alpha \in \text{spec}_-\, C_{Vir} \\ \log \rho_{\text{prim}}(h, \bar{h}) & \text{else} \end{cases}. \qquad (4.21)$$

**Semiclassicality Conditions**  To complete the analysis, we need to check that the conditions (4.15) and (4.17) actually hold in some reasonable states. I will first consider a thermofield double state and then a slightly more general class of states. For simplicity, I focus on states with no angular momentum; the discussion generalises straightforwardly.

The thermofield double at a temperature $\beta$ (and zero spin) is

$$|\beta\rangle = \frac{1}{\sqrt{Z(\beta)}} \sum_O e^{-\beta E_O} \sum_{\mathbf{m}, \bar{\mathbf{m}}} e^{-\beta(N_\mathbf{m} + N_{\bar{\mathbf{m}}})} |O, \mathbf{m}, \bar{\mathbf{m}}\rangle \otimes |O, \mathbf{m}, \bar{\mathbf{m}}\rangle. \qquad (4.22)$$

Here, $N$ is the level of the descendant. The probability distribution over primaries and the density matrix of each primary sector is

$$p_O = \frac{e^{-\beta E_O}}{\eta(\tau = i\beta/2\pi)^2 Z(\beta)}, \quad \rho_O = \eta(\tau)^2 \sum_{\mathbf{m}, \bar{\mathbf{m}}} e^{-\beta(N_\mathbf{m} + N_{\bar{\mathbf{m}}})} |\mathbf{m}, \bar{\mathbf{m}}\rangle \langle \mathbf{m}, \bar{\mathbf{m}}|, \qquad (4.23)$$

where $\eta$ is the Dedekind eta function (the thermal partition function of a generic Verma module). More precisely, the descendant density matrix is different for the identity module; but if the temperature is above the Hawking-Page transition (our main interest) the contribution of this block is negligible.

It is immediately obvious that (4.17) is satisfied, since $\rho_O$ is independent of $O$; so we only need to check (4.15). First, let us estimate $\delta p_{\alpha,O}$ as introduced in (4.11). It is well-known that the thermal distribution has $\mathcal{O}(\sqrt{c})$ width in energy (since $c$ plays the role of volume in this thermodynamic limit). So we can pretend that $p_O \sim \exp\{-a_1[E_O - E(\beta)]^2/c\}$. Within

a window of $\mathcal{O}(1)$ centred at $\alpha$, defining $\delta E_O = E_O - E_\alpha$,

$$p_O \sim \exp\left\{-a_1 \frac{[E_\alpha - E(\beta)]^2}{c} + a_2 \delta E_O \frac{E_\alpha - E(\beta)}{c}\right\}$$

$$\approx \exp\left\{-a_1 \frac{[E_\alpha - E(\beta)]^2}{c}\right\}\left(1 + a_2 \delta E_O \frac{E_\alpha - E(\beta)}{c}\right)$$

$$\implies \quad \delta p_{\alpha,O} \sim c^{-1/2}. \tag{4.24}$$

In the last equation, I have taken $E_\alpha - E(\beta) \sim \sqrt{c}$ and $\delta E_O \sim c^0$, since these are the regimes that the state has support on. Since $H(O|\alpha) - \langle \hat{A} \rangle \sim \delta p_{\alpha,O}^2$, we conclude that the difference is $\mathcal{O}(1/c)$ and the first condition in (4.15) is satisfied ($H(O) \sim c$).

As for $H(\alpha|O)$, since it is the uncertainty in the bin given the primary, we can estimate it as the log of the number of bins with support on $O$. Since each bin has $\mathcal{O}(1)$ width, the number of bins with support on a given $O$ is also $\mathcal{O}(1)$. Averaging over positive $\mathcal{O}(1)$ quantities is guaranteed to yield an $\mathcal{O}(1)$ quantity, and so we conclude that $H(\alpha|O) \sim 1 \ll H(O) \sim c$. Note that the satisfaction of this condition depends on a very coarse-grained property of the state, only that its entropy is $\mathcal{O}(c)$; its mostly a check of the coarse-graining rather than the semiclassicality of the state.

We can also consider microcanonical TFDs, in whom the width of $p_O$ is some large $\mathcal{O}(1)$ value $\sigma_O$. If the width of the bin functions is $\sigma_\alpha$, then the same considerations as (4.24) tells us that $\delta p_{\alpha,O} \sim \sigma_\alpha/\sigma_O$ and therefore that $H(O|\alpha) - \langle \hat{A} \rangle \sim (\sigma_\alpha/\sigma_O)^2$.[12] This will also be true for superpositions over $\mathcal{O}(1)$ microcanonical TFDs. These are a large class of possible semiclassical states we might be interested in.

## 4.3 Comparison to the Bulk

We have derived an information-theoretic area operator, but as mentioned in the introduction there is a logical distinction between information-theoretic and geometric areas. We need to check that they match.

The identification of the bulk area with the entropy of the primaries is commonplace by now, see e.g. [24–26, 40–48]. The essential idea is that this is the part of the entropy that can't be attributed to boundary gravitons, which are manifestly localised far away from the horizon. Since it is not new, the result that the area is the entropy of the primaries has already been checked.

It is only the derivation that is new in this work. There are some interesting consequences of the derivation presented here. For example, in the two cases in (4.21), we clearly see the two sides of the Hawking-Page transition.

Another point of interest is that we have recreated some aspects of previous work from the top down. In [24], by considering a Hamiltonian version of the Maloney-Witten trick [49] of expressing the high-energy sector in terms of the identity block in the cross-channel,

---

[12]Note that this puts an $\mathcal{O}(1)$ lower bound on the width of the state. [11] took $\sigma_\alpha$ to be a small negative power of $c$, and this possibility can also be explored if (2.1) is true for such a window.

the authors concluded that the bulk Hilbert space above the threshold had the form

$$\mathcal{H}_{\text{bulk}} = \mathcal{H}_\alpha \otimes \mathcal{H}_{\text{desc}}^{\otimes 2}. \tag{4.25}$$

What we have done with the coarse-graining is recreate this bulk Hilbert space. In (4.25), $\mathcal{H}_\alpha$ is not localised to either side, and its existence is a manifestation of the Hamiltonian factorisation problem [50]. We saw in (3.11) that the exact code did not have a factorisation problem, and here we see that getting the area operator required passing to an approximate code in which there is a factorisation problem. This connects the factorisation problem and the existence of an area operator in the way envisioned in [50].

A technical advantage in recreating this bulk story is as follows. For example, the canonical thermofield double state in $\mathcal{H}_{\text{bulk}}$ is[13]

$$|\beta\rangle_{\text{bulk}} = \int dr_h \sqrt{\rho_{\text{prim}}(r_h)} e^{-\frac{\beta}{2}\frac{r_h^2}{8G_N}} |r_h\rangle \sum_{\mathbf{n}} e^{-\frac{\beta}{2}E_{\mathbf{n}}} |\mathbf{n}\rangle |\mathbf{n}\rangle. \tag{4.26}$$

[24] propose a factorisation map (simplifying their story somewhat)

$$J_{\text{bulk}} |r_h\rangle = \frac{1}{\sqrt{\rho_{\text{prim}}(r_h)}} \int_{-\infty}^{\infty} ds |r_h, s\rangle |r_h, s\rangle. \tag{4.27}$$

When we calculate the entanglement entropy, there is a divergence that comes from the integral over $s$, and so [24–26] only find that entanglement agrees with area up to a $\log \infty$. A similar issue plagues work in JT gravity [21, 22]. In the approach presented here, the variable $s$ takes on the meaning of the individual primaries inside a microcanonical window and the $\log \infty$ is replaced by $H(\alpha|O)$.

The results in this work also naturally connect to fixed-area states [9, 10]. Fixed-area states are defined using the gravitational path integral to be states where the HRT surface in some homology/homotopy class has an area that is only allowed to fluctuate an $\mathcal{O}(G_N)$ amount. They also depend on a choice of window function [11], same as in the coarse-graining above. Thus, we are led to a statement that the fixed $\alpha$ states considered in this work are dual to fixed-area states.

## 5 Other Cases

We can perform the same analysis as in section 3.2 in a few other cases with minimal complications. The coarse-graining story is essentially parallel to that in section 4. I list them here.

### 5.1 Nearly Conformal Quantum Mechanics

The first example is nearly conformal quantum mechanics, which can be thought of as a single draw from the SSS ensemble dual to JT gravity [51]. Here, take $\mathcal{H}_{\text{code}} =$

---

[13]$r_h^2/8G_N$ is the ADM mass, and $\rho_{\text{prim}}(r_h)$ includes Jacobian factors due to the change of integration variable.

$\{f(H_L) \otimes \mathbb{1} |\epsilon\rangle\}$, where $|\epsilon\rangle$ takes the same form as in (3.1). Since $H_L - H_R |\epsilon\rangle = 0$, it follows that $H_L - H_R|_{\mathcal{H}_{\text{code}}} = 0$. Thus, $\mathcal{H}_{\text{code}} = \text{span}\{|E\rangle_L |E\rangle_R | E \in \text{spec} H\}$.

This code subspace, along with the additional restriction $E > 0$, is dual to the set of two-boundary states in pure JT gravity. Thus, in this case the area operator is given by the choices

$$
\begin{aligned}
\mathfrak{O} &= \text{spec} H \bigcap \mathbb{R}^+ \\
\alpha &\in \mathbb{R}^+ \\
f_\alpha(E) &= \text{a set of functions supported on } E > 0 \text{ with } \mathcal{O}(1) \text{ width} \\
\log n_\alpha &= S_0 + \sqrt{2\phi_r E},
\end{aligned}
\tag{5.1}
$$

where $S_0, \phi_r$ are the usual coupling constants in the bulk JT gravity. Here, there are no degrees of freedom in each $\mathsf{O}$ sector.

## 5.2 A Single Interval

Now, consider a single CFT on $S^1$, split into two complementary regions $B, \overline{B}$. One cannot define entanglement entropy of $B$ with $\overline{B}$, since the algebra of the region $B$ is type $\text{III}_1$. We will define the entropy of a specific type I approximation of the algebra of $B$ [52], using a slight modification of the construction introduced in [53].

**Factorisation map**    First, I define a factorisation map [22, 53] from the Hilbert space of the CFT on a circle to two copies of that on an interval, $J : \mathcal{H}_{S^1} \to \mathcal{H}_B^{(\epsilon)} \otimes \mathcal{H}_{\overline{B}}^{(\epsilon)}$. The factorised Hilbert spaces will depend on a Cardy boundary condition $\sigma$. I assume in this work the existence of only one such condition, with the property that $\langle \mathbb{1} \rangle_\sigma = g_\sigma \neq 0$. $g_\sigma$ is also equal to the inner product $\langle\langle\sigma|\Omega\rangle$ between the Cardy boundary state $|\sigma\rangle\rangle$ and the $S^1$ vacuum $|\Omega\rangle$.

The factorisation map I will use is

$$
J_{\epsilon,\delta} : \quad \mathcal{H}_{S^1} \to \mathcal{H}_{I(\sigma\sigma)} \otimes \mathcal{H}_{I(\sigma\sigma)},
$$

$$
J_{\epsilon,\delta} = \quad \text{} \quad \text{with} \quad \delta \ll \epsilon \ll 1.
\tag{5.2}
$$

This picture is a section of Euclidean path integral, and therefore implicitly describes an operator with the indicated domain and range. The red lines here, and below, mean a physical boundary with Cardy boundary condition $\sigma$.

While $J$ is not an isometry, it becomes proportional to an isometry in the limit $\delta, \epsilon \to 0$,

with the order of limits shown in (5.2). To see this, note that

$$\text{} = \left(\frac{\epsilon}{\delta}\right)^{\frac{c}{12}}\text{} = \left(\frac{\epsilon}{\delta}\right)^{\frac{c}{6}} g_\sigma |0\rangle + \mathcal{O}\left(e^{\frac{c}{12}-E_1}\log\epsilon/\delta\right)$$

$$\approx g_\sigma \left(\frac{\epsilon}{\delta}\right)^{\frac{c}{6}} \text{} \tag{5.3}$$

where $E_1$ is the energy of the first excited state. The prefactor in the first equality comes from the Weyl anomaly, see e.g. [54] for an explicit computation. Since the Euclidean path integral that creates the operator $J^\dagger J$ contains two copies of the annulus on the left, we can 'close up' the two holes to find a cylinder of height $2\epsilon$. In the limit $\epsilon \to 0$, the height of the cylinder with the closed up holes goes to 0 and thus we find $J^\dagger_{\delta,\epsilon} J_{\delta,\epsilon} \propto \mathbb{1}$ up to terms suppressed in the limit. From here onwards, I will drop the explicit dependence on $\epsilon, \delta$.[14]

**Code** The code in this case is

$$\mathcal{H}_{\text{code}} = J\mathcal{H}_{\mathbb{1}}, \tag{5.4}$$

where $\mathcal{H}_{\mathbb{1}}$ is the Hilbert space of descendants of the identity. Apart from the factorisation map, this is the code suggested in [18].[15]

The basis for each interval is given by

$$|L_{-\mathbf{n}}O\rangle \propto \text{}. \tag{5.5}$$

The wavefunction of a state $|\Psi\rangle \in \mathcal{H}_{\text{code}}$ in this basis is

$$\Psi[O, \mathbf{n}; O'\mathbf{n}'] \propto \langle L_{-\mathbf{n}}O \otimes L_{-\mathbf{n}'}O'|J|\Psi\rangle \propto \text{}, \tag{5.6}$$

where the path integral has been drawn in a convenient conformal frame. As before, we find that the wavefunction is non-zero only when $O = O'$ and so

$$\mathcal{H}_{\text{code}} = \bigoplus_O \mathcal{V}_O^{(B)} \otimes \mathcal{V}_O^{(\overline{B})} \tag{5.7}$$

---

[14]The two differences from the construction in [53] are as follows. First, they sum over many boundary conditions in a specific way so that the prefactor in (5.3) does not appear; since that involves an assumption about the set of boundary conditions, I drop it at the cost of the factor. This factor does not play a role, since it disappears when we normalise the state. Secondly, they take the two Cardy boundaries to have different sizes, whereas I take them to be symmetric.

These difference aside, this work is deeply indebted to [53].

[15]Indeed, [18] was the starting point for this work. The Heisenberg picture story suggested there will be explored in future work.

as before.

**Area Operator**   There is a subtlety here due to orders of limits. If we take a state of energy that diverges when $\epsilon/\delta \to \infty$, we might in principle get factorised states. However, if we keep the energy finite in this limit, we will not find this. Playing with this order of limits might give a different area operator, but I will follow the same procedure as above.

The area operator then is given by the following choices. $\mathfrak{O}$ is the spectrum of the BCFT with $\sigma\sigma$ boundary conditions (we only need consider above-threshold states in this case), labelled by a single conformal weight $\Delta$. $\alpha$ is valued in $(c/24, \infty)$. $f_\alpha(O)$ is a function peaked at $\alpha = \Delta_O$ and having $\mathcal{O}(1)$ width. The density of primaries is [33]

$$\log n_\alpha = \log \rho_{\text{prim}}(\alpha, 0) + 2 \log g_\sigma. \tag{5.8}$$

## 6   Discussion

In this work, I have shown how one can construct an area operator in a particularly simple error-correcting code in the CFT. I have also given evidence that it agrees with the geometric area. The most important aspect of this derivation was an explicit coarse-graining step, whose role is take the modular Hamiltonian (which has discrete spectrum) and convert it into an operator with continuous spectrum.

Some interesting points that came up were as follows. Firstly, it is possible to write down an area operator that knows about Hawking-Page transitions. Secondly, the analysis found a co-emergence of the area operator and a Hamiltonian factorisation problem. Thirdly, it lent meaning to an infinity in bottom-up constructions of area operators in [21, 22, 24–26].

An interesting open problem is this. The area operator in this work was constructed in terms of a Virasoro Casimir. From the bulk side, the area can be measured by a Wilson line/loop, see e.g. [55–57]. This Wilson loop can be deformed to the boundary and then becomes a path-ordered integral of the stress tensor [58]. Since it measures area, it must also be a Virasoro Casimir; the fact that it commutes with Virasoro generators can be seen by deforming the loop into the bulk, doing a large diffeomorphism and bringing it back to the boundary. It would be interesting to explicitly relate these two expressions.

One objection to this work is that the code subspace is very large. For example, when constructing an area operator with a small code subspace, [20] did not need to perform this step. While this is correct, I believe that the series of steps is of independent interest since it shows one way in which continuous spectrum operators emerge from a discrete theory.

Another important aspect is that this code is one where there are no local excitations in the bulk. This allowed us to be explicit in our construction. In fact, [59] found that from the bottom-up, there is no unique area operator without bulk local degrees of freedom; the top-down approach here does not run into the same problem. The inclusion of bulk local excitations should mean that there is no centre any more [60]; [61] has an interesting suggestion for a non-central area operator.

## Acknowledgements

I thank Suzanne Bintanja, Abhijit Gadde, Shiraz Minwalla, Rob Myers, Yuya Kusuki, Alok Laddha, K Narayan, Sridip Pal, Onkar Parrikar, Siddharth Prabhu, Manish Ramchander, Pratik Rath, Shashank Sengar, Tanmoy Sengupta and Sandip Trivedi for discussions.

This work has been presented at an informal talk in CMI, the conference "Quantum Information and Quantum Gravity 2025" held in Perimeter Institute, Canada, and in a seminar series at Tata Institute of Fundamental Research, Mumbai. I thank all three sets of organisers and audiences.

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
