# Peer review of "Emergent Area Operators in the Boundary"

_SciPost Physics_

## Round 2 · Referee Report · Anonymous (Referee 1) · 2025-12-29

Strengths

Clearly written

Weaknesses

Lacks technical novelty

Results and arguments are known in the literature

Report

The author is trying give a statistical description of the coarse-graining protocol that could lead to an emergent area operator which is nonlinear in the state. Then the author tested this statistical description in holographic 2d CFT's and found known results about area operators in that case could be reproduced. This article is clearly written modulo some confusing physical explanations of various steps in the derivation.

However, the central result-- the statistical description of the coarse-graining-- in the paper has been understood in the community. Though it might not be written down in a general form in existing literature due to its transparency and simplicity, writing a small piece of the known result down is not enough to qualify as a scientific paper. This draft is better suited for pedagogical journals like SciPost lecture notes. Thus I suggest the author to consider this type of journals instead of journals reporting novel scientific results.

Recommendation

Accept in alternative Journal (see Report)

---

## Editorial Decision

in_refereeing